# INSPIRE: Intensity and spatial information-based deformable image registration

**Johan Öfverstedt***, **Joakim Lindblad, Nataša Sladoje**

Department of Information Technology, Uppsala University, Uppsala, Sweden

* johan.ofverstedt@it.uu.se

## Abstract

We present INSPIRE, a top-performing general-purpose method for deformable image registration. INSPIRE brings distance measures which combine intensity and spatial information into an elastic B-splines-based transformation model and incorporates an inverse inconsistency penalization supporting symmetric registration performance. We introduce several theoretical and algorithmic solutions which provide high computational efficiency and thereby applicability of the proposed framework in a wide range of real scenarios. We show that INSPIRE delivers highly accurate, as well as stable and robust registration results. We evaluate the method on a 2D dataset created from retinal images, characterized by presence of networks of thin structures. Here INSPIRE exhibits excellent performance, substantially outperforming the widely used reference methods. We also evaluate INSPIRE on the Fundus Image Registration Dataset (FIRE), which consists of 134 pairs of separately acquired retinal images. INSPIRE exhibits excellent performance on the FIRE dataset, substantially outperforming several domain-specific methods. We also evaluate the method on four benchmark datasets of 3D magnetic resonance images of brains, for a total of 2088 pairwise registrations. A comparison with 17 other state-of-the-art methods reveals that INSPIRE provides the best overall performance. Code is available at github.com/MIDA-group/inspire.

## 1 Introduction

Deformable image registration is the process of finding dense correspondences between images which maximize the affinity of co-occuring structures while simultaneously preserving spatial coherence [1–3]. Numerous applications in science and technology require deformable image registration. Medical and biomedical imaging scenarios often involve acquisition of images of the same specimen from multiple views or at different times, requiring image registration for information fusion [4]. Inter-subject and atlas-to-subject correspondence maps are commonly used for comparative studies as well as for atlas-based segmentation [5]. Tracking of objects in computer vision often relies on the ability to find deformable correspondence maps [6].

To reach top performance on a particular task, an existing method for deformable image registration, see *e.g.*, [1, 6, 7], must typically be tuned or modified to fit the specific characteristics

**Citation:** Öfverstedt J, Lindblad J, Sladoje N (2023) INSPIRE: Intensity and spatial information-based deformable image registration. PLoS ONE 18(3): e0282432. https://doi.org/10.1371/journal.pone.0282432

**Data Availability Statement:** "The minimal dataset can be found at https://doi.org/10.5281/zenodo.7552884."

**Funding:** This study was supported by The Wallenberg AI, Autonomous Systems and Software

Program (WASP) AI-Math initiative (https://wasp-sweden.org/) in the form of grants awarded to JL and NS. This study was also supported by VINNOVA projects (https://www.vinnova.se/en/) in the form of grants (2017-02447; 2020-03611; 2021-01420) awarded to JL and NS, and in support of JÖ. The funders had no role in study design, data collection and analysis, decision to publish, or preparation of the manuscript.

**Competing interests:** The authors have declared that no competing interests exist.

of the application, using carefully designed feature extraction and selection as well as task optimized pre-processing [8]. Intensity-based registration, involving maximization of a similarity measure using local optimization, is one of the main approaches towards general-purpose deformable image registration [9–11], in particular in the medical and biomedical contexts. Learning-based methods have become more prominent in recent years, aiming to replace manual (engineered) tuning to the specific task with learning based such. The most successful of these methods involve training a neural network to perform the registration task in only one or a few prediction steps. Once trained, they typically offer reduced run-times [7, 12–14], however requiring representative training data and in many cases delivering lower registration accuracy than the iterative approaches [7].

We present INSPIRE, a novel state-of-the-art deformable registration method based on minimization of a distance measure which combines intensity and spatial information [15], modelling the deformations as cubic B-spline transformations. INSPIRE is symmetric, intensity interpolation-free, and robust to large displacements of image structures. To further increase utility of the method, we propose an efficient Monte Carlo algorithm for estimating the distances and gradients which substantially reduces the memory required and increases applicability of the framework to big (image) data. We also propose a gradient-weighted sampling scheme to further improve the efficiency of stochastic subsampled optimization. An open-source implementation is available at github.com/MIDA-group/inspire. Through its efficient combination of stochastic subsampled optimization and cubic B-Spline transformations, INSPIRE achieves a low run-time, while simultaneously achieving high accuracy.

We conduct three major experiments comparing the performance of INSPIRE with both intensity-based methods and widely used learning-based methods: (i) Registration to recover known transformations of 2D retinal images, as illustrated in Fig 1. Here INSPIRE substantially outperforms both the learning-based method VoxelMorph [13] and intensity-based methods [10, 11]; (ii) Registration of 134 real 2D retinal image pairs of the Fundus Image Registration Dataset (FIRE), where the method is compared against five other methods, including

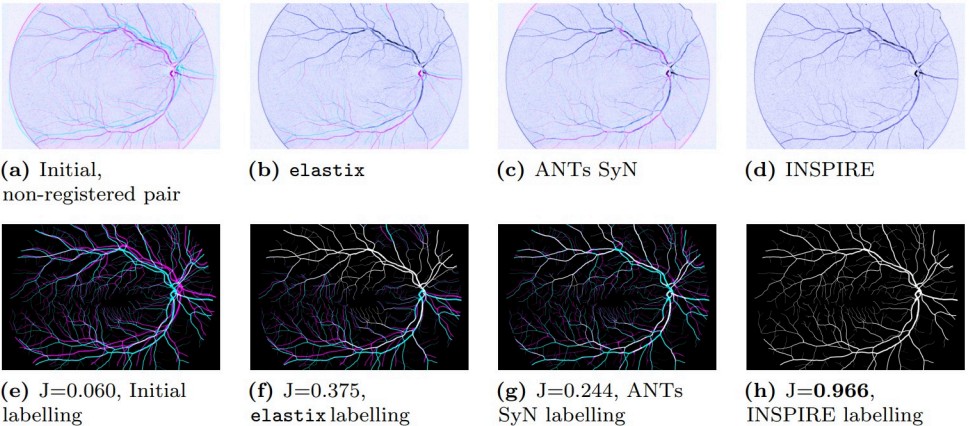

**(a)** Initial, non-registered pair   **(b)** elastix   **(c)** ANTs SyN   **(d)** INSPIRE

**(e)** J=0.060, Initial labelling   **(f)** J=0.375, elastix labelling   **(g)** J=0.244, ANTs SyN labelling   **(h)** J=**0.966**, INSPIRE labelling

**Fig 1. Example of registration of retinal images.** *Top row* (as magenta/cyan overlays of the corresponding image pair; the better the alignment of the vessel networks, the less magenta/cyan colour visible in the image): (a) Initial deformation, and the results of registration using (b) elastix [11], (c) ANTs SyN [10], and (d) INSPIRE. *Bottom row*: The respective (Ground Truth) segmented vessel networks in the observed images in the top row (only used for evaluation, not for registration), with their corresponding Jaccard index (J). We observe that elastix (b, f) registered reasonably well the upper part of the vessel network, but performed poorly on the lower region. ANTs SyN (c, g) registered some parts of the image well, but failed to obtain an accurate overall alignment, and produced severe distortions and artefacts. INSPIRE produced a very accurate alignment, with barely visible errors. Performance on the whole Retinal dataset is summarized in Fig 4.

specialized domain-specific methods. Even though INSPIRE is a general-purpose method, INSPIRE outperforms all considered methods by a large margin; (iii) Registration of inter-subject 3D magnetic resonance (MR) images of brains [16], where the method is compared against 17 other methods (including learning-based methods Quicksilver [14] and VoxelMorph [13]) on 4 datasets—LPBA40, IBSR18, CUMC12, MGH10—for a total of 2088 pairwise registrations. On IBSR18 and MGH10, INSPIRE is the best performing method, and on LPBA40 and CUMC12, INSPIRE reaches the second best performance. The method is fast, easy to use, utilizes parallel processing, and requires small amount of auxiliary memory.

## 2 Contributions, related work, preliminaries

We start with summarizing the main contributions of this paper, while also relating it to the relevant work in the field. Then we introduce the main concepts and notation that this work relies on.

### 2.1 Contributions

INSPIRE extends the affine image registration framework [15] to deformable image registration, a substantially more challenging (and ill-posed) task. The main contributions of this paper are: (i) The proposed objective function, based on the $\alpha$-AMD distance measure [15], while also incorporating an inverse inconsistency regularization term delivering symmetric registration performance also for non-invertible transformation models; (ii) A novel stochastic point-sampling method based on Gaussian gradient magnitudes that increases the performance of the optimization process; (iii) An intensity interpolation-free distance (and gradient) estimation method based on Monte Carlo integration that overcomes the limitations (high memory requirements, and requirement to quantize image intensities) of the discretization approach of [15] and enables registration of gigapixel images; (iv) A derivative normalization scheme that stabilizes the optimization and simplifies the tuning of hyperparameters; (v) A created open access image dataset of thin vessel structures, suitable for quantitative evaluation of registration methods.

### 2.2 Most relevant related work

There exist a large number of intensity-based deformable image registration methods [3, 7], out of which some are closely related to this study. In addition to methods based on B-splines [9, 17], we particularly mention two methods based on non-parametric dense displacement fields [10, 18].

Thirion's Demons [18] is a non-parametric asymmetric diffusion-based method for image registration, where a (dense) displacement field is updated via composition, using smoothing of the update field, as well as of the total field, as regularization. A variant of the Demons algorithm that addresses the lack of symmetry of the original version is the Diffeomorphic Demons method [19]. Another example of a symmetric non-parametric method is Symmetric Normalization (SyN) [10], which aims at finding two geodesic paths, one from each space (of the two images to be registered), to a common 'half-way' space. A cost term is used to maximize similarity between the two input images warped into this common space.

Among most recent work, much research has been dedicated to Deep Learning for deformable image registration. Two relevant approaches aim for general applicability: Quicksilver [14], trained to emulate an iterative method, operates on patches using a convolutional neural (encoder-decoder) network to predict a deformation model based on image appearance, imitating the "ground-truth" non-learning-based method [20] with a few prediction steps; VoxelMorph [12, 13, 21] is a successful example of another popular approach which uses

unsupervised training of a CNN to directly predict the transformation, given the full images/volumes as input. The training of the network is done with a similarity measure as objective function.

## 2.3 Preliminaries and notation

**2.3.1 Intensity-based deformable image registration.** Given a distance measure $d$ between images and a set of valid transformations $\Omega$, deformable intensity-based registration of two images, $\mathcal{A}$ (floating) and $\mathcal{B}$ (reference), can be formulated as the regularized optimization problem,

$$\hat{T} = \arg\min_{T \in \Omega} d(T(\mathcal{A}), \mathcal{B}) + R(T), \tag{1}$$

where $R$ denotes a regularization functional on $T$, modelling prior knowledge (or preference) —*e.g.*, smoothness or invertibility—of $\hat{T}$ (if not guaranteed by the choice of $\Omega$).

**2.3.2 Images as fuzzy sets.** The theory of fuzzy sets [22] provides a framework where gray-scale images are conveniently represented as spatial fuzzy sets. A *fuzzy set* $\mathcal{S}$ on a reference set $X_\mathcal{S}$ is a set of ordered pairs, $\mathcal{S} = \{(x, \mu_\mathcal{S}(x)) : x \in X_\mathcal{S}\}$. Representing images, the membership function $\mu_\mathcal{S} \colon X_\mathcal{S} \to [0, 1]$ assigns a value (intensity) to each pixel in the image domain $X_\mathcal{S}$. We denote the dimensionality of the image domain $X_\mathcal{S}$ by $n$ and, for rectangular domains, we denote the domain size by $N \in \mathbb{Z}^n$ (a vector of side lengths of the $n$D-rectangle along each dimension). The *complement* of a fuzzy set $\mathcal{S}$ is $\bar{\mathcal{S}} = \{(x, 1 - \mu_\mathcal{S}(x)) : x \in X_\mathcal{S}\}$. An *$\alpha$-cut* of a fuzzy set $\mathcal{S}$ is a crisp set ${}^\alpha\mathcal{S} = \{x \in X_\mathcal{S} : \mu_\mathcal{S}(x) \geq \alpha\}$, *i.e.*, a thresholded image. The *height $h(p)$* of a fuzzy point $p$ (a single-element fuzzy set) is equal to its intensity. For details see [15, 22, 23].

**2.3.3 Fuzzy point-to-set and set-to-set distances.** A bidirectional fuzzy point-to-set distance based on $\alpha$-cuts [23, 24], between fuzzy point $p$ and set $\mathcal{S}$ is

$$d^\alpha(p, \mathcal{S}) = \int_0^{h(p)} d(p, {}^\alpha\mathcal{S})\, d\alpha + \int_0^{1-h(p)} d(p, {}^\alpha\bar{\mathcal{S}})\, d\alpha. \tag{2}$$

Given fuzzy set $\mathcal{A}$ on reference set $X_\mathcal{A} \subset \mathbb{R}^n$, fuzzy set $\mathcal{B}$ on $X_\mathcal{B} \subset \mathbb{R}^n$, a weight function $w_A : X_\mathcal{A} \to \mathbb{R}_{\geq 0}$, and a crisp subset (mask) $M_B \subset \mathbb{R}^n$, the *Asymmetric average minimal distance for image registration* [15] from $\mathcal{A}$ to $\mathcal{B}$, parameterized by a transformation $T : X_\mathcal{A} \to \mathbb{R}^n$, is

$$d^{\;R}_{\to \alpha\text{AMD}}(\mathcal{A}, \mathcal{B}; T, w_A, M_B) =$$

$$\frac{1}{\sum_{x \in \hat{X}} w_A(x)} \sum_{x \in \hat{X}} w_A(x)\, d^\alpha(T(\mathcal{A}(x)), \mathcal{B}), \tag{3}$$

where $\hat{X} = \{x : x \in X_\mathcal{A} \wedge T(x) \in M_B\}$.

**2.3.4 B-Splines for deformable registration.** The free form deformation (FFD) model [9] is a parametric non-rigid deformation model, based on cubic B-spline transformations, that is widely used in image registration [11, 25].

Given the cubic B-spline basis functions

$$\begin{aligned} B_0(u) &= \tfrac{1}{6}(1-u)^3, \quad B_1(u) = \tfrac{1}{6}(3u^3 - 6u^2 + 4), \\ B_2(u) &= \tfrac{1}{6}(-3u^3 + 3u^2 + 3u + 1), \quad B_3(u) = \tfrac{1}{6}u^3, \end{aligned} \tag{4}$$

and a $c_1 \times \cdots \times c_n$ mesh $\Phi$ of control-points $\Phi_{i_1, \ldots, i_n} \in \mathbb{R}^n$, with dimension-wise uniform

spacing $\delta_1, \ldots, \delta_n$, positioned on a finite rectangular region (constituting its support), a B-spline transformation $T : \mathbb{R}^n \to R^n$ is given by [9]

$$T(x) = x + \sum_{i_1=0}^{3} \cdots \sum_{i_n=0}^{3} \left( \prod_{j=1}^{n} B_{i_j}(u_j) \right) \Phi_{z_1+i_1,\ldots,z_n+i_n}, \tag{5}$$

where $z_k = \lfloor \frac{x_k}{\delta_k} \rfloor - 1$ and $u_k = \frac{x_k}{\delta_k} - \lfloor \frac{x_k}{\delta_k} \rfloor$. If $x$ is outside of the finite support, $T$ is taken to be the identity transformation.

Registration based on (5) is compatible with two primary multi-scale strategies (which can be combined) for increasing both robustness and accuracy: (i) Gaussian resolution pyramids, and (ii) multi-resolution B-spline grids [9].

**2.3.5 Symmetric transformations and inverse inconsistency.** Symmetric registration frameworks, where the roles of the reference and floating images are interchangeable, generally exhibit improved robustness and performance as well as reduced impact of interpolation. Some methods are symmetric by construction [10, 19], although, the involved iterative estimation of dense displacement fields make them slow. However, not all methods nor all transformation models (cubic B-splines, *e.g.*) are invertible (or closed under inversion). Given transformation $T_{AB}$, which denotes a transformation that aligns image $A$ with image $B$, and transformation $T_{BA}$ that aligns image $B$ with image $A$, $T_{AB}$ and $T_{BA}$ may not be each other's inverses. In such cases, where a registration approach and transformation model does not provide invertibility by construction, we may still obtain many of the benefits of an invertible model through a regularization scheme.

The inverse inconsistency measure [26] quantifies the deviation of a pair of transformations $T_{AB}$ and $T_{BA}$ from being each other's inverses, and is, for a point $x$ (in the space of image $A$), given by

$$\mathrm{IIC}(T_{AB}, T_{BA}; x) = \frac{1}{2} \| T_{BA}(T_{AB}(x)) - x \|^2. \tag{6}$$

The total IIC, given by, *e.g.*, summation or averaging, can be used as a cost term to promote invertibility, an idea introduced in [26] which has remained a topic of interest in image registration [17, 27, 28].

**2.3.6 Stochastic sampling of points.** The efficiency of many intensity-based image registration methods can be dramatically increased by use of stochastic subsampling [15]. Quasi-random (or stratified) sampling shows to be a preferable choice in stochastic subsampling, compared to, *e.g.*, uniform subsampling [29]. One well performing approach is an $n$-dimensional Kronecker sequence, where the $i$-th sample $\boldsymbol{x}(i) = (x_1(i), \ldots x_n(i))$ is given by

$$x_j(i) = \lfloor (ia_j \bmod 1)N_j \rfloor, \tag{7}$$

for suitable (irrational) choices of $a_1, \ldots, a_n$. It has been observed [30] that generalized golden ratio numbers $a_1 = \phi_n^{-1}, \ldots, a_n = \phi_n^{-n}$ are suitable choices. They are given by the following nested radical recurrence

$$\phi_i = \sqrt[(i+1)]{1 + \sqrt[(i+1)]{1 + \sqrt[(i+1)]{1 + \cdots}}} \ . \tag{8}$$

## 3 Intensity and spatial information-based deformable image registration

We introduce INSPIRE—a novel method for performing deformable image registration by directly solving the optimization problem defined by (1) using local optimization. To this end, we define an objective function consisting of a distance term which combines intensity and spatial information [15, 23], and a regularization term which promotes inverse consistency [26], while we choose a set of transformations ($\Omega$) based on cubic B-splines [9]. INSPIRE addresses registration by solving a sequence of above described optimization problems (which we refer to as stages) in a coarse-to-fine manner, with images of varying resolution and smoothness, and utilizing transformations with varying number of control-points [9]. Each optimization problem is solved using local search with a gradient-based optimization procedure, starting from an interpolated version of the transformation found at the previous stage [31, 32].

### 3.1 An objective function for deformable image registration

We first define our novel objective function for image registration. It builds on distance measures of our framework for affine symmetric registration [15], but excludes the assumption of existence of (or access to) an inverse of the transformation. Aiming for symmetric image registration (even) in the absence of invertible transformations, we incorporate, as regularization, a soft constraint based on the IIC (6), inspired by [17, 26, 27]. The proposed objective function is directly compatible with cubic B-spline transformations or other spline variations, as well as registration based on non-parametric displacement fields.

Let a fuzzy set (image) $\mathcal{A}$ on reference set (domain) $X_{\mathcal{A}}$, fuzzy set $\mathcal{B}$ on $X_{\mathcal{B}}$, weight functions $w_A : X_{\mathcal{A}} \to \mathbb{R}_{\geq 0}$ and $w_B : X_{\mathcal{B}} \to \mathbb{R}_{\geq 0}$, crisp subsets (masks) $M_A, M_B \subset \mathbb{R}^n$, crisp subsets (sets of sample points) $P_A \subseteq X_{\mathcal{A}}$ and $P_B \subseteq X_{\mathcal{B}}$, and regularization parameter $\lambda \in \mathbb{R}_{\geq 0}$, be given.

We define the *average asymmetric weighted inverse inconsistency* (AW-IIC) of a transformation pair $T_{\mathrm{AB}}$, $T_{\mathrm{BA}}$ as

$$\mathrm{AW} - \mathrm{IIC}(T_{\mathrm{AB}}, T_{\mathrm{BA}}; X_{\mathcal{A}}, w_A, M_B) = \frac{1}{\sum_{x \in \hat{X}_{\mathcal{A}}} w_A(x)} \sum_{x \in \hat{X}_{\mathcal{A}}} w_A(x) \, \mathrm{IIC}(T_{\mathrm{AB}}, T_{\mathrm{BA}}; x), \qquad (9)$$

where $\hat{X}_{\mathcal{A}} = \{x : x \in X_{\mathcal{A}} \wedge T_{\mathrm{AB}}(x) \in M_B\}$.

Combining (3) and (9), we define the core of our proposed registration framework—an *objective function for deformable image registration with inverse inconsistency regularization*, for a transformation pair $T_{\mathrm{AB}}$, $T_{\mathrm{BA}}$, as

$$J(T_{\mathrm{AB}}, T_{\mathrm{BA}}; \mathcal{A}, \mathcal{B}, P_A, P_B, w_A, w_B, M_A, M_B, \lambda) =$$

$$\frac{1}{2} \big( \underset{\to \alpha \mathrm{AMD}}{d^R} (\mathcal{A} \cap P_A, \mathcal{B}; T_{\mathrm{AB}}, w_A, M_B) + \underset{\to \alpha \mathrm{AMD}}{d^R} (\mathcal{B} \cap P_B, \mathcal{A}; T_{\mathrm{BA}}, w_B, M_A) \big) + \quad (10)$$

$$\frac{\lambda}{2} \big( \mathrm{AW} - \mathrm{IIC}(T_{\mathrm{AB}}, T_{\mathrm{BA}}; P_A, w_A, M_B) + \mathrm{AW} - \mathrm{IIC}(T_{\mathrm{BA}}, T_{\mathrm{AB}}; P_B, w_B, M_A) \big).$$

Using subsets $P_A \subset X_{\mathcal{A}}$ and $P_B \subset X_{\mathcal{B}}$ created by random sampling in (10) enables using optimization methods that rely on stochastic sampling, such as stochastic gradient descent. This is a strategy to speed up parametric (B-spline) models that do not require dense sampling.

The objective function (10) suitably combines the squared norm used in (6) with the (non-squared) Euclidean point-to-point distance inbuilt in (3): for small inverse inconsistencies the

distance term dominates while for larger inverse inconsistencies the IIC term gradually overtakes.

## 3.2 Gradient-based stochastic optimization

To minimize (10), we employ *stochastic gradient descent with momentum* (SGDM), with subsets ($P_A$ and $P_B$), created by sampling, when computing the objective function and its derivatives. The gradient of the first part of (10), related to distance between the images, is given by the estimation method presented in Sec. 3.3, whereas the derivative of (6) (the second part of (10)) follows by differentiation of (5) and the chain-rule.

In the following subsections, we introduce a novel point-sampling method and a weight-normalization scheme for improving the performance of the stochastic optimization.

**3.2.1 Novel point-sampling method.** We observe that large smooth corresponding image regions tend to overlap early in the registration process, yielding very small gradients for the subsequent iterations.

*Gradient-weighted sampling of points*. We propose to sample the points (in an image domain $X$) with non-uniform probabilities given by the normalized spatial Gaussian gradient magnitude

$$p_\Delta(\mathcal{X} = x) = \frac{\mathrm{GM}(x; \sigma_{\mathrm{GM}}, t_{\mathrm{GM}})}{\sum_{y \in X} \mathrm{GM}(y; \sigma_{\mathrm{GM}}, t_{\mathrm{GM}})}, \tag{11}$$

where $\mathcal{X}$ is the random point variable and $\mathrm{GM}(y; \sigma_{\mathrm{GM}}, t_{\mathrm{GM}})$ denotes the Gaussian gradient magnitude, with smoothing $\sigma_{\mathrm{GM}}$ and where values below $t_{\mathrm{GM}}$ are set to 0. If no points exceed the gradient magnitude threshold, (11) is undefined, in which case we use uniform sampling instead. The parameter $\sigma_{\mathrm{GM}}$ of the Gaussian-derivative allows tuning to the size of structures and the desired width of the region of mass around edges. An example of sampling based on (11), applied to a retinal image, is shown in Fig 2.

In comparison to previous approaches [33], this sampling does not depend on the transformation; each image is sampled independent of the other and the probabilities thus remain unchanged for all iterations corresponding to a given resolution level in the coarse-to-fine pyramid optimization, saving computational time. Furthermore, [33] is asymmetric, assigning distinct roles to the two images, unlike the proposed sampling method.

*Combination of sampling schemes*. Sampling from $p_\Delta$ (11) is more computationally demanding than uniform sampling ($p_{\mathbb{U}}$). Furthermore, it omits uniform regions entirely, which can

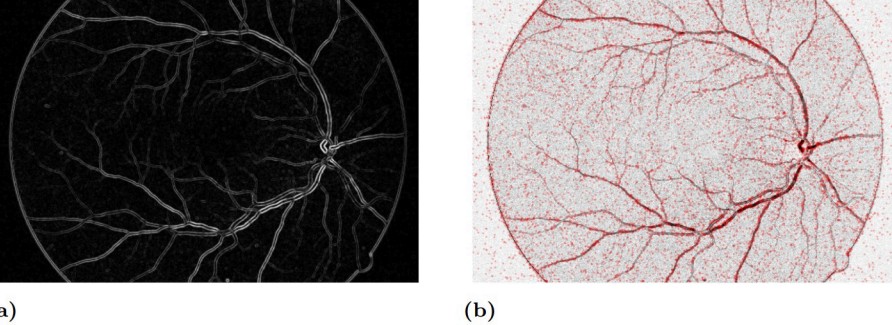

(a)                                         (b)

**Fig 2. Gradient-weighted stochastic sampling of points on a retinal image.** (a) Probability mass function (11) with $\sigma_{\mathrm{GM}} = 5.0$ (normalized); (b) 10000 sample points overlaid on the image as red semi-transparent markers.

result in failed registration or locally high inverse inconsistency, if used exclusively. We therefore propose a sampling method which combines the two methods to obtain the benefits of both $p_\Delta$ and $p_\mathbb{U}$ without their respective drawbacks. The suggested mixture probability model (parameterized by $m$) is

$$p_m(\mathcal{X} = x) = mp_\Delta(\mathcal{X} = x) + (1 - m)p_\mathbb{U}(\mathcal{X} = x)\,, \tag{12}$$

where $p_\Delta$ is the gradient-weighted sampling method (11) and $p_\mathbb{U}$ is the uniform sampling method. First a sampler is selected at random, and then a point is sampled from the chosen sampler.

**3.2.2 Derivative scaling.**   The combination of the SGDM optimizer and cubic B-spline transformations poses a particular challenge. The magnitudes of the derivatives related to a given transformation parameter depend heavily on grid coarseness and how many (and which) points are sampled inside the parameter's local support. To both decrease the impact of the random sampling on the magnitude of the derivatives and to simplify tuning of the stepsize (by making it less dependent of grid coarseness), we propose a derivative scaling scheme for the partial derivatives of the objective function $J$.

Let $T^i$ denote the $i$-th parameter of the transformation $T$, i.e, one coordinate of one of the control-points in $\Phi$ (5). The partial derivatives w.r.t. $T^i$ are scaled in the SGDM optimization by the factor $\frac{1}{\gamma^i_{AB} + \varepsilon}$, where $\gamma^i_{AB}$ expresses the total impact of the parameter on the transformation of the points in $P_A$ and $P_B$ (in (10)), and is given by

$$\gamma^i_{AB} = \sum_{x \in \hat{P}_A} w_A(x) \left\| \frac{\partial T_{AB}(x)}{\partial T^i_{AB}} \right\| +$$
$$\lambda \sum_{x \in \hat{P}_A} w_A(x) \left\| \frac{\partial (T_{BA}(T_{AB}(x)))}{\partial T^i_{AB}} \right\| + \tag{13}$$
$$\lambda \sum_{x \in \hat{P}_B} w_B(x) \left\| \frac{\partial (T_{AB}(T_{BA}(x)))}{\partial T^i_{AB}} \right\|,$$

whereas $\varepsilon$ is a small constant (we use $\varepsilon = 0.01$).

## 3.3 Estimation of $\alpha$-cut distances and their spatial derivatives by Monte Carlo sampling

Computation of the objective function (10) and its derivatives requires estimation of (3), which is based on the point-to-set distance measure (2). The family of methods proposed in [15, 23] for performing this estimation on rectangular (grid) domains require quantization of (image) intensities into a discrete number of levels, followed by computation of distance transforms on each intensity level. This approach has two main limitations: (i) loss of information due to (coarse) quantization, (ii) substantial run-time and memory usage for computation and storage of the distance transforms. Here we propose a novel method for estimating (2) with an approximation based on Monte Carlo integration. Given $N_\alpha$ randomly sampled $\alpha$-cuts, for $\alpha_1, \ldots, \alpha_{N_\alpha} \in [0, 1]$, the estimator $\widetilde{d}(p, \mathcal{S}) \approx d^\alpha(p, \mathcal{S})$ is defined as

$$\widetilde{d}(p, \mathcal{S}) = \frac{1}{N_\alpha} \sum_{i=1}^{N_\alpha} d'(p, \mathcal{S}; \alpha_i)\,, \tag{14}$$

where

$$d'(p, S; \alpha) = \begin{cases} d(p,^{\alpha} \mathcal{S}), & \text{for } \alpha \leq h(p), \\ d(p,^{(1-\alpha)}\bar{\mathcal{S}}), & \text{for } \alpha > h(p). \end{cases} \qquad (15)$$

**Algorithm 1** Monte Carlo-based Distance and Gradient Estimation: Main Procedure

**Input:** Fuzzy point $p$ and set $\mathcal{S}$, trees $\tau(\mathcal{S})$, $\tau(\bar{\mathcal{S}})$.

**Output:** Distance estimate $\widetilde{d}(p, \mathcal{S})$ of $d^{\alpha}(p, \mathcal{S})$ and the discrete estimate $\widetilde{g}(p, \mathcal{S})$ of its gradient at $p$.

```
1. procedure MC − DG(p, 𝒮; τ(𝒮), τ(𝒮̄), s)
2:    P ← NEAREST-GRID-POINTS(p)
3:    for j ∈ {1, ..., N_α} do
4:      α ← SAMPLE(0, 1)
5:      if α ≤ h(p) then              ▷ Inwards point-to-set distance
6:        D_j ← SEARCH(1, 0, N, d_MAX; P, α, τ(𝒮))
7:      else                          ▷ Complement distance
8:        D_j ← SEARCH(1, 0, N, d_MAX; P, 1 − α, τ(𝒮̄))
9:      end if
10:   end for
11:   D̃ = (1/N_α) ∑_{j=1}^{n} D_j
12:   d̃ ← INTERPOLATE(p, D̃)
13:   g̃ ← DISCRETE − GRADIENT(MID − POINT(P), D̃)
14:   return d̃, g̃
15: end procedure
```

**Algorithm 2** Monte Carlo-based Distance and Gradient Estimation: Search Procedure

**Require:** Node $i$, rectangle $(y, R)$, current (smallest) distances $D$, evaluation points $\mathbf{P}$, level $\alpha$, tree $\tau$.

**Ensure:** Updated distances by evaluating the sub-tree $\tau_i$.

```
1: procedure SEARCH(i, y, R, D; P, α, τ)
2:    if α > τ_i or ⋀_{j∈{1...2ⁿ}} D_j ≤ d_R(P_j, y, R; s) then
3:      return D
4:    end if
5:    if ∏_{j=1}^{n} R_j = 1 then
6:      return min(D_1, d(P_1, y)), ..., min(D_{2ⁿ}, d(P_{2ⁿ}, y))
7:    end if
8:    y_(1), y_(2), R_(1), R_(2), k ← SPLIT-RECT(y, R; s)
9:    if [P_1]_k ≤ [y_(2)]_k then
10:     D ← SEARCH(LEFT(i), y_(1), R_(1), D; P, α, τ)
11:     D ← SEARCH(RIGHT(i), y_(2), R_(2), D; P, α, τ)
12:   else
13:     D ← SEARCH(RIGHT(i), y_(2), R_(2), D; P, α, τ)
14:     D ← SEARCH(LEFT(i), y_(1), R_(1), D; P, α, τ)
15:   end if
16:   return D
17: end procedure
```

Algorithm 1 provides an efficient way to compute $\widetilde{d}(p, \mathcal{S})$, relying on a suitable data structure constructed by Alg. 3. It overcomes the listed limitations of previous methods. No intensity quantization, nor intensity interpolation are required, and only very light-weight pre-processing is performed.

**Algorithm 3** Construction of Data-Structure for Monte Carlo based estimation of point-to-set Distance and Gradient

```
Input: Fuzzy set 𝒮.
Output: KD-tree τ(𝒮).
1: procedure BUILD-TREE(𝒮)
```
2:  $\quad \tau(\mathcal{S}) \leftarrow \text{NEW} - \text{ARRAY}(1 \ldots 2^{1 + \sum_{j=1}^{n} \lceil \log_2(R_j) \rceil})$
3:  $\quad \text{BUILD} - \text{TREE} - \text{REC}(\tau(\mathcal{S}), 1, \mathbf{0}, N; \mathcal{S})$
```
4:     return τ(𝒮)
5: end procedure
6: procedure BUILD-TREE-REC(τ, i, y, R; 𝒮)
```
7:  $\quad$ **if** $\prod_{j=1}^{n} R_j = 1$ **then**
8:  $\quad\quad \tau_i \leftarrow \mu_{\mathcal{S}}(y)$
```
9:     else
```
10: $\quad\quad y_{(1)}, \; y_{(2)}, \; R_{(1)}, \; R_{(2)}, \; k \leftarrow \text{SPLIT-RECT}(y, R; s)$
11: $\quad\quad \text{BUILD} - \text{TREE} - \text{REC}(\tau, \text{LEFT}(i), y_{(1)}, R_{(1)}; \mathcal{S})$
12: $\quad\quad \text{BUILD} - \text{TREE} - \text{REC}(\tau, \text{RIGHT}(i), y_{(2)}, R_{(2)}; \mathcal{S})$
13: $\quad\quad \tau_i \leftarrow \max(\tau_{\text{LEFT}(i)}, \; \tau_{\text{RIGHT}(i)})$
```
14:    end if
15: end procedure
```

Algorithm 3 constructs an augmented KD-tree [34], $\tau(\mathcal{S})$, that corresponds to the input image $\mathcal{S}$. Within each node of $\tau(\mathcal{S})$ we store the maximum intensity of that sub-tree. This enables an efficient distance search, where sub-trees which are empty after $\alpha$-cutting can be pruned. The discrete rectangular domain enables storing the node attributes compactly in a single array. The left and right sub-trees of a node indexed by $i$ have indices $2i$ and $2i + 1$ respectively, where the root has index 1.

**Sampling and multi-sample search.** An initial step of Algorithm 1 is to sample $\alpha$-cuts for which to compute the distance. We propose to use the quasi-random sampling (7), with $n = 1$. Algorithm 1 can be implemented to perform a joint search for the closest point distances for all the $N_\alpha$ sampled $\alpha$-cuts in parallel, yielding a substantially lower computational time. To simplify the pseudo-code, multi-sample search is omitted from the description (but is used in our provided implementation).

**Sub-pixel distances.** To enable evaluation of (14) everywhere in the image space, we compute the distance from all nearby grid-points of the point of interest $p$, followed by distance interpolation. We use bilinear/trilinear interpolation to compute the sub-pixel distance, and the discrete gradient approximation evaluated in the mid-point (to avoid vanishing derivatives at saddle-points of the interpolation surface).

**Space subdivision and lower distance bound.** To facilitate efficient search for closest point distances in Alg. 1, using the (implicit) KD-tree data structure constructed by using Alg. 3, we require systematic methods for space subdivision and for computing lower distance bounds from points to rectangles corresponding to nodes in the tree.

Given an $n$D-rectangle of size $R = (R_1, \ldots, R_n)$ spels ($n$D-pixels), with $y$ denoting its corner point with minimal coordinates, and $s = (s_1, \ldots, s_k, \ldots, s_n)$ the spel spacing along each dimension, let SPLIT-RECT($y$, $R$;$s$) be a function that splits a rectangle in half (up to integer precision), into two integer-sized sub-rectangles, along the dimension given by $\arg \max_{k \in \{1 \ldots n\}} s_k(R_k - 1)$, breaking ties systematically.

As a lower bound for the distance $d(p, {}^{\alpha}\mathcal{S})$, we use the Euclidean distance from the point $p$ to the nearest spel inside an $n$D-rectangle (characterized by $y$, $R$ and $s$) with maximum intensity $\geq \alpha$, given by

$$d_R(p, y, R; s) = \sqrt{\sum_{i=1}^{n} \max(0, y_i - p_i, p_i - (y_i + s_i(R_i - 1)))^2}. \tag{16}$$

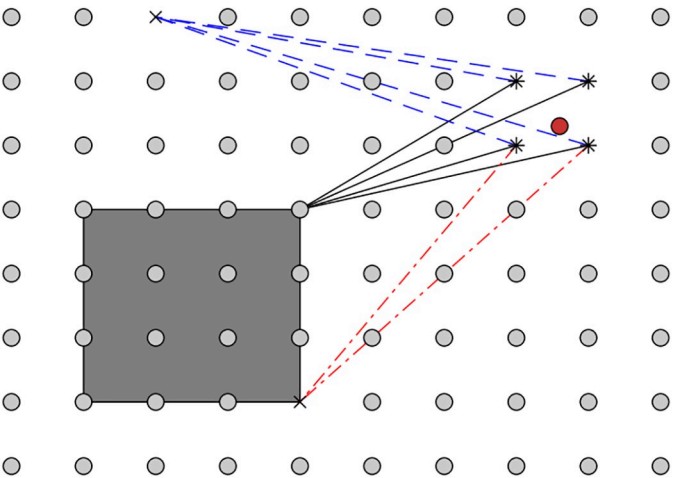

**Fig 3. Example of the search process in Alg. 1.** The red circle is the fuzzy point $p$, from which the distance is computed, and the four nearby star ($^*$) markers indicate the nearest grid points ($\mathbf{P}$). The two crosses ($\times$) represent points of $\mathcal{S}$ that surpass the sampled intensity threshold $\alpha$. The gray rectangle shows the region $(y, R)$ corresponding to the current node $i$ in the KD-tree. The (lengths of the) dashed lines (blue) represent the distances from each of the nearest grid-points to the current closest point (the upper $\times$). The (lengths of the) solid lines (black) represent the lower bound of the distance to any point within the rectangle. In the example, the lower bound is smaller than at least one of the current lowest distances, and the rectangle contains at least one object point, and thus the sub-tree will be searched. The (lengths of the) dash-dotted (red) lines represent the distances from the grid-points for which a closer point was found by searching within the rectangle.

To further speed up the search, we may substitute $d_R$ with $\hat{d}_R$, which relaxes the bound by a factor of $\beta$ for distance bounds which exceed a provided threshold $d_t$. This may significantly reduce the number of nodes that need to be searched in exchange for potential over-estimation of far (thus, likely less relevant) distances. $\hat{d}_R$ is given by

$$\hat{d}_R(p, y, R; s, d_t, \beta) = \max(0, \beta[d_R(p, y, R; s) - d_t]) + \min(d_t, d_R(p, y, R; s)), \tag{17}$$

where $\beta \in \mathbb{R}_{\geq 1}$, $d_t \in [0, d_{\text{MAX}}]$.

Fig 3 illustrates an ongoing search process.

### 3.4 Preprocessing and implementation

INSPIRE uses Gaussian resolution pyramids and a coarse-to-fine pyramid B-spline schedule with increasing number of control points. At each level, a downsampling factor, a parameter $\sigma$ controlling the width of the Gaussian filter used to smoothen the image, and the number of control-points (along each dimension) are to be selected. Furthermore, at each level, (optional) pre-processing steps of (robust) normalization (to $[0, 1]$) using the $q$-percentile of image intensities $\mu_\mathcal{S}$ (similar as in [15]), and histogram equalization, are used to remove differences in value range (and outliers), and homogenize the global contrast of the images to be registered.

INSPIRE is implemented in the C++ language, as an extension of the Insight Toolkit (ITK) [35].

## 4 Performance analysis

To validate the practical relevance of the proposed method, we present results of an empirical study of the performance of INSPIRE, compared with several state-of-the-art methods, when

used to solve image registration tasks in 2D and 3D. The data and several scripts to generate the plots for the paper can be located in [36].

## 4.1 Synthetic evaluation on retinal images

Thin structures, such as vessel networks appearing in medical images, present a challenge for registration methods. To assess the performance of the proposed method on deformable registration of images containing thin structures with varying thickness and contrast, we construct a synthetic evaluation task, using real medical (retinal) images. Available annotated ground-truth masks of the vessel networks enable quantitative performance assessment based on comparison of the degree of mask overlap after registration.

Most existing deformable registration methods rely on similarity measures that are based on differences in intensity of overlapping points. Such similarity measures are potentially ill-suited for thin structures, where large transformations are typically handled by use of Gaussian pyramids with a high degree of smoothing, a strategy that risks erasing the structures of interest. INSPIRE, on the other hand, can deal with large transformations without much smoothing.

**4.1.1 Construction of the dataset.**   We create a new dataset suitable for evaluation of image registration methods, with focus on thin structures of varying thickness and contrast. Starting from the High Resolution Fundus Image Database [37], which consists of 42 images of size $3504 \times 2336$, we perform the following sequence of processing steps:

- The images are converted from RGB to grayscale;

- Background variation is removed with a rolling ball of radius 20 pixels, to remove low frequency features which may be exploited by a registration method;

- The images are padded (with 350 px on each side);

- The images are deformed in a coarse-to-fine sequence of random elastic transformations (B-splines with 7, 14, 24, 48 control points, with random perturbations to each parameter of $(-250, 250)$, $(-50, 50)$, $(-25, 25)$, $(-15, 15)$ at each level respectively, creating 42 pairs of "floating" (deformed) and "reference" (non-deformed) images.

The last two image pairs, 41 and 42, are designated training images. The created dataset is accessible on the Zenodo platform [38]. An example image pair is shown in Fig 1(a).

**4.1.2 Experiment setup.**   In this study, INSPIRE is benchmarked against widely used general purpose registration methods, the parametric registration toolkit `elastix` [11], and the non-parametric ANTs SyN [16]. We also evaluated a feature-based method based on SIFT-features [39] (using a variety of settings), but the obtained feature point-sets were consistently too noisy to be useful, so we excluded that approach from the full study. Finally, we also compare with the recent learning-based method VoxelMorph, both without (VM) [13, 21] and with (100 steps of) instance optimization (VM-IO).

**4.1.3 Method configuration.**   Parameter tuning of all the methods were performed on the training image pairs 41 and 42. The main parameters tuned for all the methods are: number of pyramid levels and step-lengths (or maximum gradient magnitude for `elastix`); additionally for ANTs SyN: field regularization, and for INSPIRE: inverse inconsistency factor $\lambda$ and gradient-weighted sampling parameters $m$ and $\sigma_{GM}$. The parameters chosen for INSPIRE are listed in Table 1.

For `elastix`, we used mean square difference as distance measure, a 6 level Guassian pyramid, a B-spline transformation model with final grid spacing of 50 pixels, adaptive stochastic optimizer, 3000 iterations per level and 40000 random spatial samples per iteration.

**Table 1. INSPIRE parameter configuration for evaluation on retinal images.** For all levels: $\lambda = 0.005$, $N_\alpha = 7$, normalization percentile $q = 0.025\%$ and histogram equalization disabled. KD-tree search approximation constants are chosen as $\beta = 1.2$ and $d_t = 20$.

| Level | Subsampling factor | $\sigma$ | Control-points | Iterations | Sampling fraction | $\frac{d_{MAX}}{diameter}$ | Step-size | Momentum | GW ($m$, $\sigma_{GM}$) |
|---|---|---|---|---|---|---|---|---|---|
| 1 | 4 | 5 | 10 | 3000 | 0.004 | 0.2 | 20 | 0.9 | 0.5, 1 |
| 2 | 2 | 3 | 24 | 2000 | 0.004 | 0.2 | 12 | 0.25 | 0.5, 2 |
| 3 | 2 | 1 | 48 | 1000 | 0.004 | 0.1 | 8 | 0.25 | 0.5, 3 |
| 4 | 1 | 0 | 64 | 250 | 0.004 | 0.1 | 3 | 0.25 | 0.5, 5 |
| 5 | 1 | 0 | 80 | 250 | 0.004 | 0.1 | 3 | 0.25 | 0.5, 5 |
| 6 | 1 | 0 | 128 | 250 | 0.004 | 0.1 | 3 | 0.25 | 0.5, 5 |

For ANTs SyN, we used mean square difference as distance measure (outperforming a correlation measure [10]), a 5 level Gaussian pyramid with sigmas (9, 5, 3, 1, 0) and downsampling factors (16, 8, 4, 2, 1). The number of iterations at each level were (300, 300, 200, 100, 100). We used an update field regularization of 5 and no total field regularization.

For VoxelMorph, we configured it using the default settings with MSD as the distance measure, which is appropriate due to the the registration task being monomodal, and trained until convergence which occurred after 20000 iterations. Both the plain version (denoted VM) [21], as well as the version of the method that includes (100 steps of) instance optimization (denoted VM-IO) [13] are included. Since the images did not fit into the GPU, we performed both training and inference on sub-patches of 512 × 512 pixels with 50% overlap in each direction (yielding a 1024 × 1024 image), such that no deformation is larger than what can fit into this overlap. Furthermore, we split the dataset in two folds, where fold-1 comprises the first 19 images and fold-2 comprises the last 19 images, and select the reference image for half of the images in each fold and the floating image for the other half.

**4.1.4 Results.** The results of the registrations are summarized in Fig 4. INSPIRE exhibits excellent results on all images, while all of the reference methods fail to exhibit stable and good performance. VoxelMorph in particular is struggling to align the thin vessels. This is likely

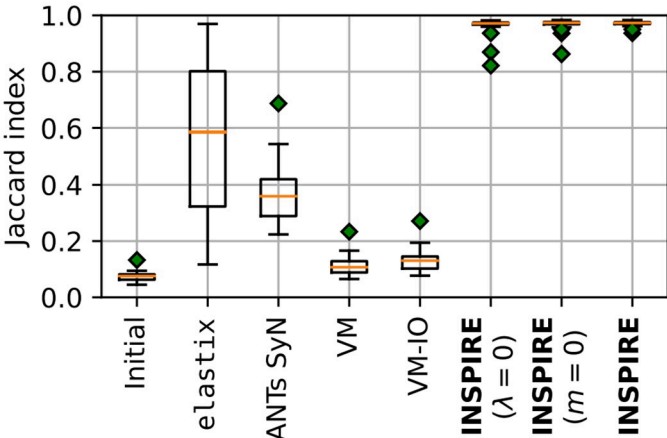

**Fig 4. Box-and-whiskers plot of Jaccard index on the Retinal dataset for the considered methods.** A higher score is better. INSPIRE exhibits very good performance for all cases, while the competing methods are much less reliable and with substantially lower performance on average. Empirical mean and std-dev of the Jaccard index computed on the ground-truth vessel masks for respective method are: (Initial, $J = 0.074 \pm 0.016$), (elastix, $J = 0.565 \pm 0.270$), (ANTs SyN, $J = 0.368 \pm 0.099$), (VoxelMorph (VM) $J = 0.112 \pm 0.032$), (VM-IO $J = 0.129 \pm 0.038$), (INSPIRE ($\lambda = 0$), $J = 0.965 \pm 0.029$), (INSPIRE ($m = 0$), $J = 0.966 \pm 0.026$), (INSPIRE, $J = 0.972 \pm 0.008$). The results corresponding to image 2 in the dataset are shown in Fig 1.

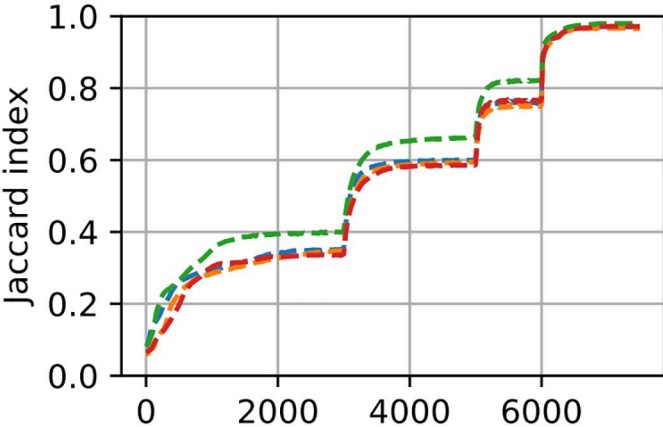

**Fig 5. Accuracy, in terms of the Jaccard index, as a function of the number of iterations for the first four retinal image registrations (shown in different colours) with INSPIRE.** The points of steep increase in accuracy correspond to the changes of coarseness-level allowing more freedom to optimize to local deformations. Especially in the later levels, most of the progress happens in rather few iterations.

due to the lack of overlap between the corresponding structures in the reference and floating images and therefore correspondences need to be established over large distances; even though the corresponding structures are within each other's receptive field (for the chosen U-net architecture), the registration is unsuccessful.

As an ablation study, we also observed INSPIRE with $\lambda = 0$, which corresponds to disabling the inverse inconsistency regularization, as well as INSPIRE with $m = 0$, which corresponds to turning off GW sampling. Both features show to contribute to improved performance and robustness.

In addition to the quantitative results presented here, we include an illustrative animation of an image registration process with INSPIRE on image 4 in the dataset (shown in Fig 1), enabling a qualitative assessment of the method, included as supplementary material in S1 File.

**4.1.5 Accuracy and number of iterations.** In image registration, there is typically a trade-off between the robustness and quality of the results, and the time required to obtain them. We have quantified this trade-off observing the registration of a subset (first 4) of the observed retinal images. At every 10-th iteration of the registration we output the registered ground-truth mask and compute the Jaccard index. The resulting graph, shown in Fig 5, indicates that if speed is of a higher priority than accuracy, there is a lot of room for reducing the number of iterations. See also Sec. 4.4.

## 4.2 Image registration of real retinal image pairs

Having observed excellent performance on the synthetic retinal image dataset, we proceed to evaluate the proposed method on real retinal image pairs, and use the open dataset *Fundus Image Registration Dataset (FIRE)* [40]. The FIRE dataset consists of 134 retinal colour image pairs to be registered, each pair consisting of two distinct image acquisitions of the same retina.

**4.2.1 Preprocessing, rigid registration and method configuration.** Before we can perform deformable registration of the images with INSPIRE, a few preprocessing steps are required.

**Table 2. INSPIRE parameter configuration for evaluation on retinal images from the FIRE dataset.** For all levels: $\lambda = 0.01$, $N_\alpha = 31$, normalization percentile $q = 0.1\%$ and histogram equalization disabled. KD-tree search approximation constants are chosen as $\beta = 1.2$ and $d_t = 20$.

| Level | Subsampling factor | $\sigma$ | Control-points | Iterations | Sampling fraction | $\frac{d_{MAX}}{\text{diameter}}$ | Step-size | Momentum | GW ($m$, $\sigma_{GM}$) |
|-------|--------------------|----------|----------------|------------|-------------------|------------------------------------|-----------|----------|-------------------------|
| 1 | 4 | 0 | 12 | 500 | 0.05 | 0.05 | 5 | 0.3 | 0.5, 1 |
| 2 | 4 | 0 | 18 | 500 | 0.05 | 0.05 | 5 | 0.3 | 0.5, 2 |
| 3 | 4 | 0 | 24 | 500 | 0.03 | 0.03 | 5 | 0.3 | 0.5, 3 |
| 4 | 2 | 0 | 28 | 500 | 0.02 | 0.02 | 3 | 0.3 | 0.5, 5 |
| 5 | 1 | 0 | 32 | 50 | 0.02 | 0.01 | 2 | 0.3 | 0.5, 5 |
| 6 | 1 | 0 | 40 | 50 | 0.02 | 0.01 | 1 | 0.3 | 0.5, 5 |

- (i) We remove the colour from the images through averaging of the three colour channels;

- (ii) We remove the bias field by subtracting a blurred image (Gaussian blur with $\sigma = 10$) from the original image;

- (iii) We perform a robust normalization of the images (only considering the contents within the mask) with percentiles chosen as (0.01 and 0.75) which eliminates some bright artifacts which can disturb the registration.

After the preprocessing we perform an initial rigid registration to find a coarse alignment before performing the deformable registration. Given the small overlaps in the P subset, we perform a global rigid alignment by computing the $d_{\alpha AMD}$ (3) for all discrete translations, which can be computed efficiently using the Fast Fourier Transform (FFT) algorithm [23]. This step is repeated for 32 rotation angles [−10, 10] discretized in a grid, and implementing the use of masks in a similar way to the process described in [41]. The rigid registration is performed on 4×-downsampled images (along each dimension) for efficiency reasons, and with the requirement of at least 40% overlap of the masks to filter out rigid transformations where the overlapping region contains no vessel-structures. The full preprocessing and rigid registration implementation is included in [36].

The parameters chosen for INSPIRE (the deformable registration) are listed in Table 2.

**4.2.2 Experiment setup.**   We apply the standardized evaluation procedure related to the FIRE dataset [40] by comparing the mean Euclidean distance between ground-truth landmarks provided by an expert annotator in the reference space after registration. Finally, we examine the results in terms of success-rate, i.e., the fraction of successful registrations out of all performed registrations, as a function of a success-threshold of the mean Euclidean landmark distance.

We compare with the following five domain-specific methods: REMPE [42], HM-16 [43], Harris-PIIFD [44], GDB-ICP [45], and SuperRetina [46].

**4.2.3 Results.**   We present the results of the experiment in Fig 6. INSPIRE outperforms the domain-specific methods substantially, being both more robust as well as more accurate in comparison with the considered methods. The proposed approach failed on the rigid registration for a single image pair (from the P subset) which had a very small overlap.

## 4.3 Deformable image registration of 3D MR images of brains

A common task used to assess the performance of deformable image registration methods is registration of manually labelled 3D MR images of brains, using the target overlap (measured in the reference image space) of each distinct label as the quality measure [5, 14].

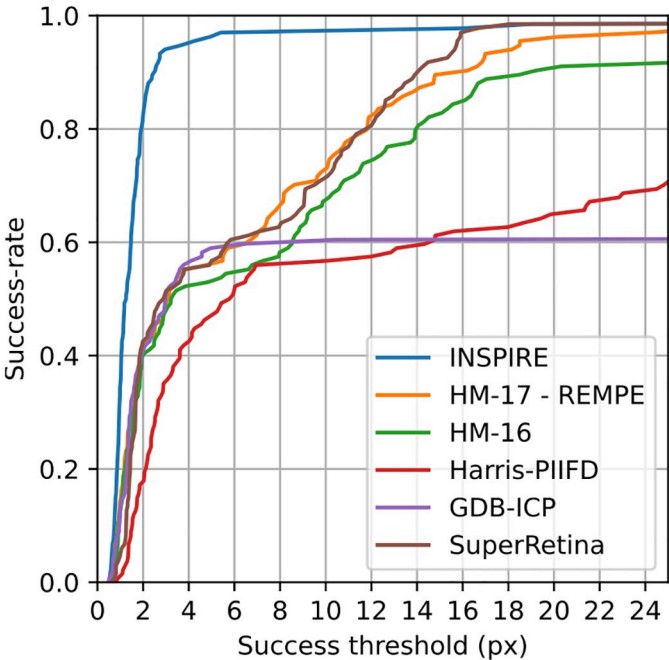

**Fig 6. Results of retinal image registration on the FIRE dataset.** The *x*-axis represents a range of error thresholds in pixels and the *y*-axis represents the fraction of registrations succeeding at each threshold level, where a registration is considered a success if the error (measured as the mean Euclidean distance between the ground-truth landmarks after registration) is below that threshold. We observe that INSPIRE exhibits excellent performance; 80% of the registrations have an error of 2 px or less, and only one single registration failed completely.

**4.3.1 Experiment setup.** We consider four public datasets in this evaluation, LPBA40 [47], IBSR18, CUMC12, and MGH10, consisting of 40, 18, 12 and 10 images respectively. Within each dataset, each image is registered to all other, giving a total of 2088 pairwise registrations. We follow the evaluation protocol of [5], thereby enabling direct comparison with 17 state-of-the-art methods, including two deep learning-based registration methods, Quicksilver [14] and VoxelMorph [13]. Following [5], each image is first affinely registered to the ICBM MNI152 non-linear atlas [48] using *NiftyReg* [25], such that each image is of size $229 \times 193 \times 193$, except for LPBA40 where the images are of size $229 \times 193 \times 229$.

**4.3.2 Method configuration.** The chosen configuration of INSPIRE, described in Table 3, is based on the configuration used in Sec. 4.1, with a few changes on account of the smaller image diameter to (i) make the optimization more stable and (ii) speed up the method.

**Table 3. INSPIRE parameter configuration for subject-to-subject registration of 3D MR images of brains.** For all levels: $\lambda = 0.01$, $N_\alpha = 7$, normalization percentile $q = 0.1\%$, histogram equalization (using 256 bins) is enabled, $\beta = 1.2$ and $d_t = 20$.

| Level | Subsampling factor | $\sigma$ | Control-points | Iterations | Sampling fraction | $\frac{d_{MAX}}{diameter}$ | Step-size | Momentum | GW ($m$, $\sigma_{GM}$) |
|-------|-------|-------|-------|-------|-------|-------|-------|-------|-------|
| 1 | 1 | 4 | 9 | 1000 | 0.005 | 0.2 | 3 | 0.9 | 0.5, 2 |
| 2 | 1 | 4 | 14 | 1000 | 0.005 | 0.2 | 3 | 0.9 | 0.5, 2 |
| 3 | 1 | 4 | 24 | 1000 | 0.005 | 0.1 | 2 | 0.3 | 0.5, 2 |
| 4 | 1 | 4 | 48 | 750 | 0.01 | 0.05 | 1 | 0.3 | 0.5, 2 |
| 5 | 1 | 4 | 64 | 300 | 0.02 | 0.025 | 1 | 0.3 | 0.5, 2 |
| 6 | 1 | 4 | 80 | 100 | 0.02 | 0.025 | 1 | 0.3 | 0.5, 2 |

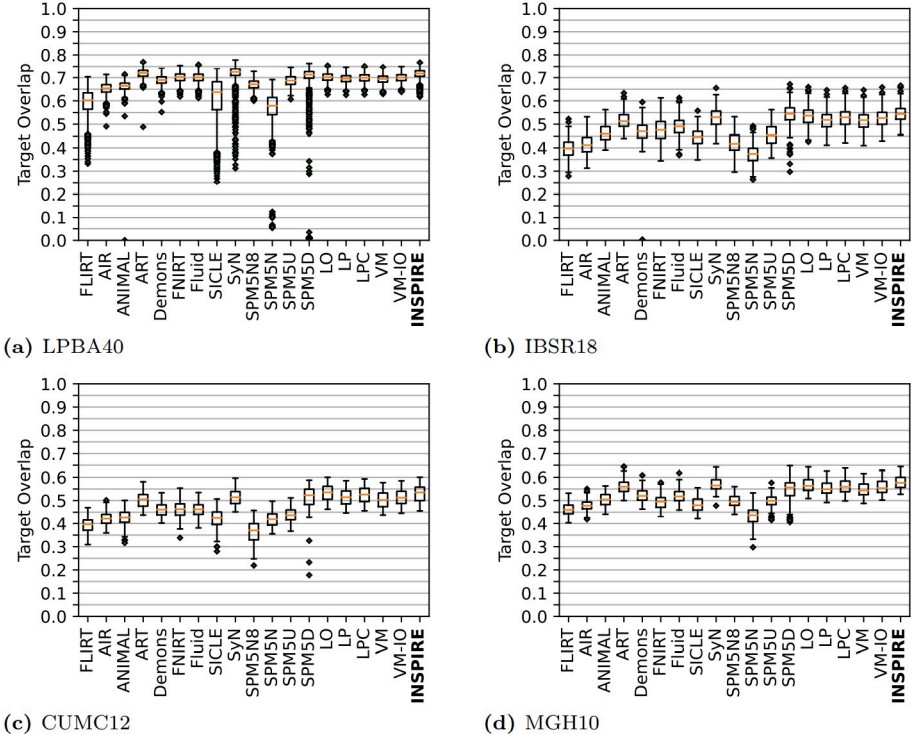

**Fig 7. Target overlap (averaged over all labels) shown as box-and-whiskers plots for the four MR datasets.** A higher score is better. The center line of the box marks the median, bottom and top mark the first and third quartiles $(Q_1, Q_3)$, and bottom and top of the whiskers mark the minimum and maximum (excluding outliers beyond $Q_1 - 1.5$ $(Q_3 - Q_1)$, and $Q_3 + 1.5(Q_3 - Q_1)$, which are shown as diamond markers). The proposed method, INSPIRE, is bolded. See [14] for details of the experiment protocol and the compared methods.

Histogram equalization was enabled due to the difference in contrast of the different volumes, which was not sufficiently addressed by normalization. Based on prior experience [15], we omit downsampling and smoothing of the (already fairly low-resolution) 3D MR brain volumes. Configuration of the reference methods are detailed in [5, 14]. We also included the recent and widely used unsupervised learning-based method VoxelMorph [13], which is configured using the default settings with MSD as the distance measure, which performed well in the original study [13] on similar data, and trained until convergence—which occurred after 20000 iterations. Both the plain version (denoted VM) [21], as well as the version of the method that includes (100 steps of) instance optimization (denoted VM-IO) [13] are included in our evaluation.

**4.3.3 Results.** The performance of INSPIRE, in comparison to the performance of 17 other deformable registration methods, on the 4 considered datasets of MR brain images [5, 14], is summarized in Fig 7. FLIRT denotes the output of an established affine registration method included for reference. All the registrations start from pre-processed images obtained (together with tables of results) through correspondence with the authors of [14]. The results are presented as box-and-whiskers plots which visualize the shape of the full empirical distribution of the results over all registrations for each method, with one plot per dataset. The top 7 ranked methods for each dataset w.r.t. mean average target overlap are shown in Table 4. INSPIRE is the top ranked method on the IBSR18 and MGH10 datasets, and the second best on the LPBA40 and CUMC12 datasets.

**Table 4. Top 7 rankings of the compared methods (see [14]) w.r.t. the empirical mean of target overlap for the four datasets of brain volumes.** Shown are empirical mean and std-dev of the target overlap. Lower rank is better and higher overlap value is better.

|  | RANK 1 | RANK 2 | RANK 3 | RANK 4 | RANK 5 | RANK 6 | RANK 7 |
|---|---|---|---|---|---|---|---|
| LPBA40 | ART | **INSPIRE** | SyN | LO | FNIRT | Fluid | VM-IO |
|  | 0.7185 ± 0.019 | 0.7156 ± 0.021 | 0.7146 ± 0.051 | 0.7015 ± 0.019 | 0.7008 ± 0.020 | 0.7002 ± 0.021 | 0.6992 ± 0.019 |
| IBSR18 | **INSPIRE** | SPM5D | LO | SyN | LPC | VM-IO | ART |
|  | 0.5450 ± 0.035 | 0.5418 ± 0.047 | 0.5331 ± 0.041 | 0.5281 ± 0.042 | 0.5258 ± 0.042 | 0.5249 ± 0.039 | 0.5154 ± 0.035 |
| CUMC12 | LO | **INSPIRE** | LPC | LP | SyN | VM-IO | SPM5D |
|  | 0.5341 ± 0.035 | 0.5287 ± 0.036 | 0.5254 ± 0.034 | 0.5146 ± 0.034 | 0.5138 ± 0.033 | 0.5134 ± 0.035 | 0.5120 ± 0.056 |
| MGH10 | **INSPIRE** | SyN | LO | ART | LPC | VM-IO | LP |
|  | 0.5785 ± 0.027 | 0.5683 ± 0.029 | 0.5665 ± 0.029 | 0.5611 ± 0.030 | 0.5607 ± 0.029 | 0.5563 ± 0.030 | 0.5531 ± 0.029 |

## 4.4 Run-time

**4.4.1 Experiment setup.** Deformable image registration is a challenging task [5, 13], where some commonly used methods can require several minutes or even hours of processing time; methods using direct optimization of (dense) displacement fields tend to have especially high run-time, while model-based methods using B-Spline transformations are substantially faster, and learning-based methods tend to be the fastest type of methods. Here we evaluate the empirical run-time of INSPIRE, elastix, ANTs SyN, as well as VoxelMorph.

We run INSPIRE and VoxelMorph, with 4 repetitions, on (i) a retinal image pair (image pair 1), and (ii) a brain image pair (MGH10, images 1 and 2), and measure the mean and std-dev of the run-time, when using 14 worker threads on a 3.3 GHz Intel(R) Core(TM) i9–9940X processor with 14 physical cores, 19.25 MB cache, 96 GB RAM, and a NVidia GeForce RTX 2080 Ti GPU (used by VoxelMorph).

**4.4.2 Results.** The run-times for both the retinal and the brain data are shown in Table 5. As can be seen for the comparison on the retinal data, INSPIRE exhibits efficient run-time in comparison to other non learning-based methods while achieving high accuracy. VoxelMorph is substantially faster, while on the other hand, achieving very much lower (substandard) accuracy on the retinal images and lower accuracy on the brain images. The time to train a Voxel-Morph model for the retinal dataset was 3062s and the time to train a VoxelMorph model for the brain data was 16000s.

**Table 5. Run time of the methods for 2D and 3D registration.**

| Run time: Retinal (2D − 4204 × 3036) | |
|---|---|
| Method | Registration time |
| **INSPIRE** | 42.0 ± 0.41s |
| elastix | 190.9 ± 0.29s |
| ANTs SyN | 737.9 ± 2.35s |
| VoxelMorph (VM) | 11.89 ± 0.30s |
| Run time: Brain (3D − 229 × 193 × 192) | |
| Method | Registration time |
| **INSPIRE** | 567.2 ± 2.35s |
| VoxelMorph (VM) | 4.63 ± 0.03s |
| VoxelMorph (VM-IO) | 107.0 ± 0.46s |

### 4.5 Ethics statement

The *High Resolution Fundus Image Database*, which the synthetically deformed dataset is based on, consists of a publicly available and anonymized dataset of retinal images. This is a retrospective study and only anonymized data is used, thus ethical approval is not required.

The FIRE dataset consists of 134 images from 39 male and female patients who provided informed consent for the analysis as stated [40]: "Written informed consent was obtained before data acquisition and processing." Since we only perform a retrospective study on these images for the stated purpose, ethical approval was not required.

The 3D study has been performed retrospectively on four datasets of skull-stripped and anonymized brain volume images used soley to evaluate performance of registration methods, a study which requires no ethical approval.

## 5 Discussion

INSPIRE is highly configurable, and therefore includes a number of hyper-parameters, just like most registration frameworks; new tasks may require some tuning to reach optimal performance. We observe that the parameters which are main candidates for tuning are the resolution pyramid settings (number of levels, smoothing $\sigma$, subsampling factor, number of control-points), normalization and histogram equalization (depending on image intensity distributions), and optimizer step-size (depending on the size of the images and of the deformations). The remaining parameters usually do not require careful tuning.

INSPIRE is a mono-modal registration method; it is based on a distance measure that assumes similar corresponding intensities for good performance. This work has, accordingly, only considered applications where differences in the distribution of intensities could be mitigated using normalization and histogram equalization. INSPIRE supports multi-channel images by marginal computation (and aggregation) of the distances and gradients (per channel separately), optionally utilizing [49].

## 6 Conclusion

We have presented a new deformable image registration framework, INSPIRE. By integrating an inverse inconsistency constraint into an existing asymmetric distance measure, we formulated an objective function which enables finding (approximately) invertible transformations between images, even when the chosen transformation model is not invertible or closed under inversion. By introducing a novel method for estimating the objective function and its derivatives, based on Monte Carlo sampling, even very large images can be registered without requiring much memory, and without quantization of the image intensities. A proposed gradient-weighted sampling scheme enables efficient sampling, leading to increased accuracy for a given computational cost.

INSPIRE exhibits excellent performance on deformable registration of synthetically deformed retinal images, where the structures of interest are thin and sparse, while other compared methods fail to solve the task satisfactorily. INSPIRE dramatically outperforms recent learning-based method VoxelMorph on the retinal image registration task. In a further study using the FIRE dataset, consisting of the registration of a set of 134 pairs of retinal images, we observe that INSPIRE exhibits outstanding performance, substantially outperforming all evaluated methods, consisting of several which are designed specifically for the application domain. Our evaluation of INSPIRE on a benchmark of pairwise inter-subject registration on 4 datasets, enables its direct comparison with 17 existing methods. INSPIRE demonstrates the overall best performance, ranking first on two datasets and second on the other two. INSPIRE

outperforms the considered learning-based approaches on all 4 datasets, without requiring any training (data).

## Supporting information

**S1 File. Illustrative animation of INSPIRE applied to a retinal image pair.** Magenta/cyan representations of the intensity images (the better the alignment, the less magenta/cyan colour is visible in the image), binary vessel masks (only for evaluation purposes, not used for registration), and the deformation field displayed as a chessboard pattern are included, as well as the objective function $J$ of the registration plotted as a function of the iteration number. (GIF)

## Author Contributions

**Conceptualization:** Johan Öfverstedt, Joakim Lindblad, Nataša Sladoje.

**Data curation:** Johan Öfverstedt.

**Funding acquisition:** Joakim Lindblad, Nataša Sladoje.

**Investigation:** Johan Öfverstedt.

**Methodology:** Johan Öfverstedt, Joakim Lindblad, Nataša Sladoje.

**Project administration:** Joakim Lindblad, Nataša Sladoje.

**Resources:** Joakim Lindblad, Nataša Sladoje.

**Software:** Johan Öfverstedt.

**Supervision:** Joakim Lindblad, Nataša Sladoje.

**Validation:** Johan Öfverstedt.

**Visualization:** Johan Öfverstedt.

**Writing – original draft:** Johan Öfverstedt.

**Writing – review & editing:** Joakim Lindblad, Nataša Sladoje.

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
