## [Decision Letter · Decision Letter 0]

2 Jan 2023

PONE-D-22-32958INSPIRE: Intensity and spatial information-based deformable image registrationPLOS ONE

Dear Dr. Öfverstedt,

Thank you for submitting your manuscript to PLOS ONE. After careful consideration, we feel that it has merit but does not fully meet PLOS ONE’s publication criteria as it currently stands. Therefore, we invite you to submit a revised version of the manuscript that addresses the points raised during the review process. The manuscript had been reviewed by 2 reviewers. Both of the reviewers were of the view that your manuscript describe technically piece of scientific research. However, they have made certian comments/suggestions to further improve your work. After consideration of comments of  both reviewers, my decision is "major revision". Please incorporate comments raised by reviewers.

We look forward to receiving your revised manuscript.

Kind regards,

Gulistan Raja

Academic Editor

PLOS ONE

Journal Requirements:

2. For studies reporting research involving human participants, PLOS ONE requires authors to confirm that this specific study was reviewed and approved by an institutional review board (ethics committee) before the study began. Please provide the specific name of the ethics committee/IRB that approved your study, or explain why you did not seek approval in this case.

Reviewers' comments:

Reviewer's Responses to Questions

**Comments to the Author**

1. Is the manuscript technically sound, and do the data support the conclusions?

Reviewer #1: Yes

Reviewer #2: Yes

2. Has the statistical analysis been performed appropriately and rigorously? 

Reviewer #1: Yes

Reviewer #2: Yes

3. Have the authors made all data underlying the findings in their manuscript fully available?

Reviewer #1: Yes

Reviewer #2: Yes

4. Is the manuscript presented in an intelligible fashion and written in standard English?

Reviewer #1: Yes

Reviewer #2: Yes

5. Review Comments to the Author

Reviewer #1: This is a very nice paper on intensity and geometry based deformable registration of images. The main advantage of the contribution is the general-purpose nature of the proposed method for deformable image registration. The theoretical and algorithmic solutions aim at computational efficiency in two scenarios. The method on a 2D dataset created from retinal images, characterised by presence of networks of thin structures.The authors provide the source code of their implementation.

I find that the literature is sufficiently cited, the method is technically sound and clearly formulated. The English also reads very well.

I only the following relatively minor issues:

- In the experiments with the retinal images, authors had to resort into synthetically deforming the images, in order to simulate deformable distortions. However, open datasets for registration of retinal images, that exhibit these properties, already exist (e.g. the FIRE dataset).

- There is no visual demonstration of the method. Of course the quantitative evaluation of the approach is a significant and valuable benchmark. Nevertheless, it does not provide a qualitative demonstration of the effect of the deformable transformation in the resultant images. The images provided in Figures 1 & 2 are rather unclear and do not suffice for such a demonstration.

Reviewer #2: The authors present a method for deformable image registration, combining intensity and spatial information, utilizing elastic B-splines and incorporating inverse inconsistency penalization. Such method can be utilized for performing both 2D and 3D registration.

The introduction and related work sections provide enough relevant information to locate the proposed work within the current state of the art. They also provide good background about the math behind key elements of the method

The method section provides a very detail and supported description of the method

Evaluation section does an adequate job, providing metrics and comparisons both for 2D and 3D registrations. My biggest gripe here is that for the 2D registration, the authors perform registration of retinal images, but they create their own synthetic dataset instead of utilizing the publicly available FIRE dataset, which is widely utilized in this domain and that would also allow for easy comparison with domain specific registration methods, not only with general methods. Analysis of performance for 3D registration on brain MRI scans i good

Dataset:

- FIRE: Fundus Image Registration dataset

- https://projects.ics.forth.gr/cvrl/fire/

- https://doi.org/10.35119/maio.v1i4.42

Some interesting methods utilizing the FIRE dataset that could be used to compare with

- Semi-supervised Keypoint Detector and Descriptor for Retinal Image Matching https://doi.org/10.1007/978-3-031-19803-8_35

- Retinal fundus image registration framework using Bayesian integration and asymmetric Gaussian mixture model https://doi.org/10.1002/ima.22789

- Color fundus image registration using a learning-based domain-specific landmark detection methodology https://doi.org/10.1016/j.compbiomed.2021.105101

- REMPE: Registration of Retinal Images Through Eye Modelling and Pose Estimation https://doi.org/10.1109/JBHI.2020.2984483

- Unsupervised Deep Learning Network for Deformable Fundus Image Registration https://doi.org/10.1109/ICASSP43922.2022.9747686

Discussion and conclusion are succinct, but complete

In conclusion, this is an interesting work, with a very detailed manuscript. I would only request the authors to provide results for retinal image registration utilizing the mentioned dataset, to enable comparing their method with domain specific methods.

6. PLOS authors have the option to publish the peer review history of their article (what does this mean?). If published, this will include your full peer review and any attached files.

Reviewer #1: No

Reviewer #2: No

---

## [Author Response · Author response to Decision Letter 0]

24 Jan 2023

Respected Dr. Raja, and reviewers 1 and 2,

We thank you for your comments and suggestions regarding the submitted manuscript PONE-D-22-32958 (INSPIRE: Intensity and spatial information-based deformable image registration).

Here we respond point-by-point to the concerns and suggestions raised. The added text in the manuscript has been annotated by a blue font in the version of the manuscript that tracks changes.

Points raised by the editor

Answer to point 1:

We have looked over the instructions and changed the file extension of the figures to “.tif”, from “.png” to adhere to the requirements.

2. For studies reporting research involving human participants, PLOS ONE requires authors to confirm that this specific study was reviewed and approved by an institutional review board (ethics committee) before the study began. Please provide the specific name of the ethics committee/IRB that approved your study, or explain why you did not seek approval in this case.

Answer to point 2 and 3:

The High Resolution Fundus Image Database, which the synthetically deformed dataset is based on, consists of a publicly available and anonymized dataset of retinal images. This is a retrospective study and only anonymized data is used, thus ethical approval is not required.

The FIRE dataset consists of 134 images from 39 male and female patients who provided informed consent for the analysis: ``Written informed consent was obtained before data acquisition and processing.'' Since we only perform a retrospective study on these images for the stated purpose, ethical approval was not required.

The 3D study has been performed retrospectively on four datasets of skull-stripped and anonymized brain volume images used soley to evaluate performance of registration methods, a study which requires no ethical approval.

This statement has been added to the manuscript as 4.4. and added to the Ethics Statement related to the submission.

Answer to point 4:

The minimal dataset can be found at https://doi.org/10.5281/zenodo.7552884 which we now reference in the manuscript.

Points related to qualitative results

Reviewer 1:

There is no visual demonstration of the method. Of course the quantitative evaluation of the approach is a significant and valuable benchmark. Nevertheless, it does not provide a qualitative demonstration of the effect of the deformable transformation in the resultant images. The images provided in Figures 1 & 2 are rather unclear and do not suffice for such a demonstration.

Answer:

Thank you for raising this point. In response to this concern, we have included an additional supplementary animation media file as S1, which displays a full registration process of a pair of synthetically deformed retinal images, including both the intensity images overlapping, the vessel masks overlapping (only for evaluation purposes) and a deformed chessboard pattern which illustrates how the space is deformed. We also include a plot where the value of the objective function is plotted as a function of the iteration number. We hope that this will provide more insight into how the method operates and what the final deformations are like.

Points related to the FIRE dataset

Reviewer 1:

In the experiments with the retinal images, authors had to resort into synthetically deforming the images, in order to simulate deformable distortions. However, open datasets for registration of retinal images, that exhibit these properties, already exist (e.g. the FIRE dataset).

Reviewer 2:

Evaluation section does an adequate job, providing metrics and comparisons both for 2D and 3D registrations. My biggest gripe here is that for the 2D registration, the authors perform registration of retinal images, but they create their own synthetic dataset instead of utilizing the publicly available FIRE dataset, which is widely utilized in this domain and that would also allow for easy comparison with domain specific registration methods, not only with general methods. Analysis of performance for 3D registration on brain MRI scans i good.

Answer:

Thank you both for raising this very relevant point. The reason that we used a synthetically deformed dataset was to enable benchmarking of only the deformable component of the transformation, without an affine component which different methods may resolve in different ways. This also enabled use of a very detailed ground-truth mask for an equally detailed evaluation. It is, however, very useful to also evaluate our method on this real dataset (FIRE), constituting a more realistic setup. In our revised version of the manuscript, we have included an evaluation of the proposed method (using a few preprocessing steps to transform the images into a suitable form for INSPIRE) on the FIRE dataset, yielding very good performance. We compare INSPIRE to several existing (domain-specific methods) and conclude that INSPIRE outperforms them all. This part of the empirical study has been added as a new subsection in Section 4. The abstract, introduction, conclusion and the reference list has also changed as a consequence of these changes.

We hope that these answers and changes to the manuscript answers all your concerns to your satisfaction.

With best regards,

 Johan Öfverstedt, Joakim Lindblad, and Nataša Sladoje

---

## [Decision Letter · Decision Letter 1]

15 Feb 2023

INSPIRE: Intensity and spatial information-based deformable image registration

PONE-D-22-32958R1

Dear Dr. Öfverstedt,

We’re pleased to inform you that your manuscript has been judged scientifically suitable for publication and will be formally accepted for publication once it meets all outstanding technical requirements.

Kind regards,

Gulistan Raja

Academic Editor

PLOS ONE

Additional Editor Comments (optional):

Reviewers' comments:

Reviewer's Responses to Questions

**Comments to the Author**

1. If the authors have adequately addressed your comments raised in a previous round of review and you feel that this manuscript is now acceptable for publication, you may indicate that here to bypass the “Comments to the Author” section, enter your conflict of interest statement in the “Confidential to Editor” section, and submit your "Accept" recommendation.

Reviewer #1: All comments have been addressed

Reviewer #2: All comments have been addressed

2. Is the manuscript technically sound, and do the data support the conclusions?

Reviewer #1: Yes

Reviewer #2: Yes

3. Has the statistical analysis been performed appropriately and rigorously? 

Reviewer #1: Yes

Reviewer #2: Yes

4. Have the authors made all data underlying the findings in their manuscript fully available?

Reviewer #1: Yes

Reviewer #2: Yes

5. Is the manuscript presented in an intelligible fashion and written in standard English?

Reviewer #1: Yes

Reviewer #2: Yes

6. Review Comments to the Author

Reviewer #1: I find that the reviewers sufficiently addresses both comments raised by the reviewers and the paper merits publication.

Reviewer #2: (No Response)

7. PLOS authors have the option to publish the peer review history of their article (what does this mean?). If published, this will include your full peer review and any attached files.

Reviewer #1: No

Reviewer #2: No

---

## [Editor Report · Acceptance letter]

23 Feb 2023

PONE-D-22-32958R1 

INSPIRE: Intensity and spatial information-based deformable image registration 

Dear Dr. Öfverstedt:

I'm pleased to inform you that your manuscript has been deemed suitable for publication in PLOS ONE. Congratulations! Your manuscript is now with our production department. 

Kind regards, 

on behalf of

Dr. Gulistan Raja 

Academic Editor

PLOS ONE